# Neutralizing gut-derived lipopolysaccharide as a novel therapeutic strategy for severe leptospirosis

Xufeng Xie[1], Xi Chen[1], Shilei Zhang[1], Jiuxi Liu[1], Wenlong Zhang[1]*, Yongguo Cao[1,2]*

[1]Department of Clinical Veterinary Medicine, College of Veterinary Medicine, Jilin University, Jilin, China; [2]State Key Laboratory for Diagnosis and Treatment of Severe Zoonotic Infectious Diseases, Key Laboratory for Zoonosis Research of the Ministry of Education, Institute of Zoonosis, and College of Veterinary Medicine, Jilin University, Changchun, China

**\*For correspondence:**
zwenlong123@126.com (WZ);
ygcao82@jlu.edu.cn (YC)

**Competing interest:** The authors declare that no competing interests exist.

**Abstract** Leptospirosis is an emerging infectious disease caused by pathogenic *Leptospira* spp. Humans and some mammals can develop severe forms of leptospirosis accompanied by a dysregulated inflammatory response, which often results in death. The gut microbiota has been increasingly recognized as a vital element in systemic health. However, the precise role of the gut microbiota in severe leptospirosis is still unknown. Here, we aimed to explore the function and potential mechanisms of the gut microbiota in a hamster model of severe leptospirosis. Our study showed that leptospires were able to multiply in the intestine, cause pathological injury, and induce intestinal and systemic inflammatory responses. 16S rRNA gene sequencing analysis revealed that *Leptospira* infection changed the composition of the gut microbiota of hamsters with an expansion of Proteobacteria. In addition, gut barrier permeability was increased after infection, as reflected by a decrease in the expression of tight junctions. Translocated Proteobacteria were found in the intestinal epithelium of moribund hamsters, as determined by fluorescence in situ hybridization, with elevated lipopolysaccharide (LPS) levels in the serum. Moreover, gut microbiota depletion reduced the survival time, increased the leptospiral load, and promoted the expression of proinflammatory cytokines after *Leptospira* infection. Intriguingly, fecal filtration and serum from moribund hamsters both increased the transcription of *TNF-α*, *IL-1β*, *IL-10*, and *TLR4* in macrophages compared with those from uninfected hamsters. These stimulating activities were inhibited by LPS neutralization using polymyxin B. Based on our findings, we identified an LPS neutralization therapy that significantly improved the survival rates in severe leptospirosis when used in combination with antibiotic therapy or polyclonal antibody therapy. In conclusion, our study not only uncovers the role of the gut microbiota in severe leptospirosis but also provides a therapeutic strategy for severe leptospirosis.

## eLife assessment

The gut microbiota influences many infectious diseases; however, its role in *Leptospirosis* remains unclear. In this **fundamental** work, Xie et al. use a hamster model to show that *Leptospira* infection leads to gut pathology, an altered gut microbiota, and increased translocation. A combined use of antibiotics and LPS neutralization prolonged survival, providing a potential new therapeutic approach. This study utilizes **compelling** methods to provide new insights into this emerging disease, which could be dissected further in future studies aimed at gaining mechanistic insight and assessing the translational relevance of these discoveries.

## Introduction

Leptospirosis is a worldwide emerging zoonotic disease caused by *Leptospira* spp. (*Coburn et al., 2021*). *Leptospira interrogans* impacts approximately 1 million people, resulting in 60,000 deaths every year (*Coburn et al., 2021*). In human leptospirosis, the clinical manifestations can range from mild illness to multiorgan failure, characterized by jaundice, kidney, and liver failure, and even death (*Haake and Levett, 2015*; *Maia et al., 2022*). Currently, one important challenge is that leptospirosis is often misdiagnosed as other febrile diseases, and once the optimal treatment period is missed, leptospirosis quickly develops into the severe form, which often ends in death (*Coutinho et al., 2014*). However, antibiotic treatment is often useless in severe leptospirosis and even aggravates it, called the Jarisch–Herxheimer Reaction (*Guerrier and D'Ortenzio, 2013*). Thus, it is imperative to explore precision medicine strategies for the treatment of severe leptospirosis.

Ninety percent of infected humans may resolve spontaneously while 10% of patients are designated as susceptible hosts (*Cagliero et al., 2018*). Intestinal bleeding is a common but neglected symptom of severe leptospirosis in humans and hamster (*Alventosa Mateu, 2017*). A healthy gut environment, shaped by the presence of a healthy functional microbial ecosystem, is fundamental to instruct the immune system toward homeostasis. The intestine harbors a densely populated resident microbial community, which consists of bacteria, viruses, and fungi, called microbiota (*Kamada et al., 2013*). Gut dysbiosis has been linked to increased susceptibility to disseminated bacterial infections and sepsis in humans. In addition, studies of patients in intensive care units (ICUs) have found that dysbiosis increases the risk of nosocomial infections, sepsis, and mortality (*McDonald et al., 2020*). Our previous study proved that the gut microbiota contributed to the defense against *Leptospira* infection in mice, a resistant model of leptospirosis, without obvious intestinal lesions or increased intestinal permeability (*Xie et al., 2022*), while diarrhea was a common symptom in the late leptospirosis of hamster. As is known, the gut microbiota is closely related to the homeostasis of the intestine (*Lee et al., 2022*). A previous study demonstrated that the variation of the gut microbiota mediated the different susceptibility to cholera infection (*Alavi et al., 2020*). Focusing on the role of the gut microbiota in leptospirosis may help explain the differences in susceptibility to *Leptospira* infection in humans and develop targeted therapy for acute leptospirosis. However, whether a relationship exists between the gut microbiota and severe leptospirosis is still unclear.

In this study, we used the hamster model of leptospirosis to explore the functional role of the gut microbiota in acute leptospirosis. We provide evidence that *Leptospira* infection disrupted intestinal homeostasis, and targeting gut-derived lipopolysaccharide (LPS) improved the survival rates of hamsters with severe leptospirosis. Our study provides a basis for the development of novel therapeutic strategies to control severe leptospirosis.

## Results

### *L. interrogans* proliferated in the intestine, impaired the intestinal structure, and induced proinflammatory response

Recently, Inamasu et al. reported that subcutaneous infection with leptospires led to intestinal dissemination (*Inamasu et al., 2022*). We wondered whether different challenge routes affected the dissemination of *Leptospira* into the intestine. Thus, we compared the common intraperitoneal challenge with subcutaneous challenge in a hamster model of leptospirosis. Our study indicated that subcutaneous challenge and intraperitoneal challenge both led to leptospiral proliferation in the ileum and colon (*Figure 1—figure supplement 1A*, *Figure 1B*). There were approximately seven times more leptospires in the colon than in the ileums at articulo mortis (AM) post infection (p.i.) after intraperitoneal challenge (*Figure 1B*). Hamsters infected by intraperitoneal challenge had shortened survival time compared with hamsters infected by subcutaneous challenge (*Figure 1—figure supplement 1B*). *L. interrogans* infection disrupted the intestinal structure, caused bleeding, and led to the infiltration of inflammatory cells (*Figure 1C*). In addition, *L. interrogans* infection promoted the inflammatory response in the colons as determined by the upregulated gene expression of *lipocalin 2* (*Lcn2*)/*neutrophil gelatinase-associated lipocalin* (*NGAL*), a marker of intestinal inflammation (*Kitamoto et al., 2020*), TNF-α, IL-1β, Nos2, and TLR4 (a well-known receptor of LPS that is vital against *Leptospira* infection [*Viriyakosol et al., 2006*] and the downregulated gene expression of *IL-10*) (*Figure 1D–I*). We wondered whether proinflammatory gene expression was also related to

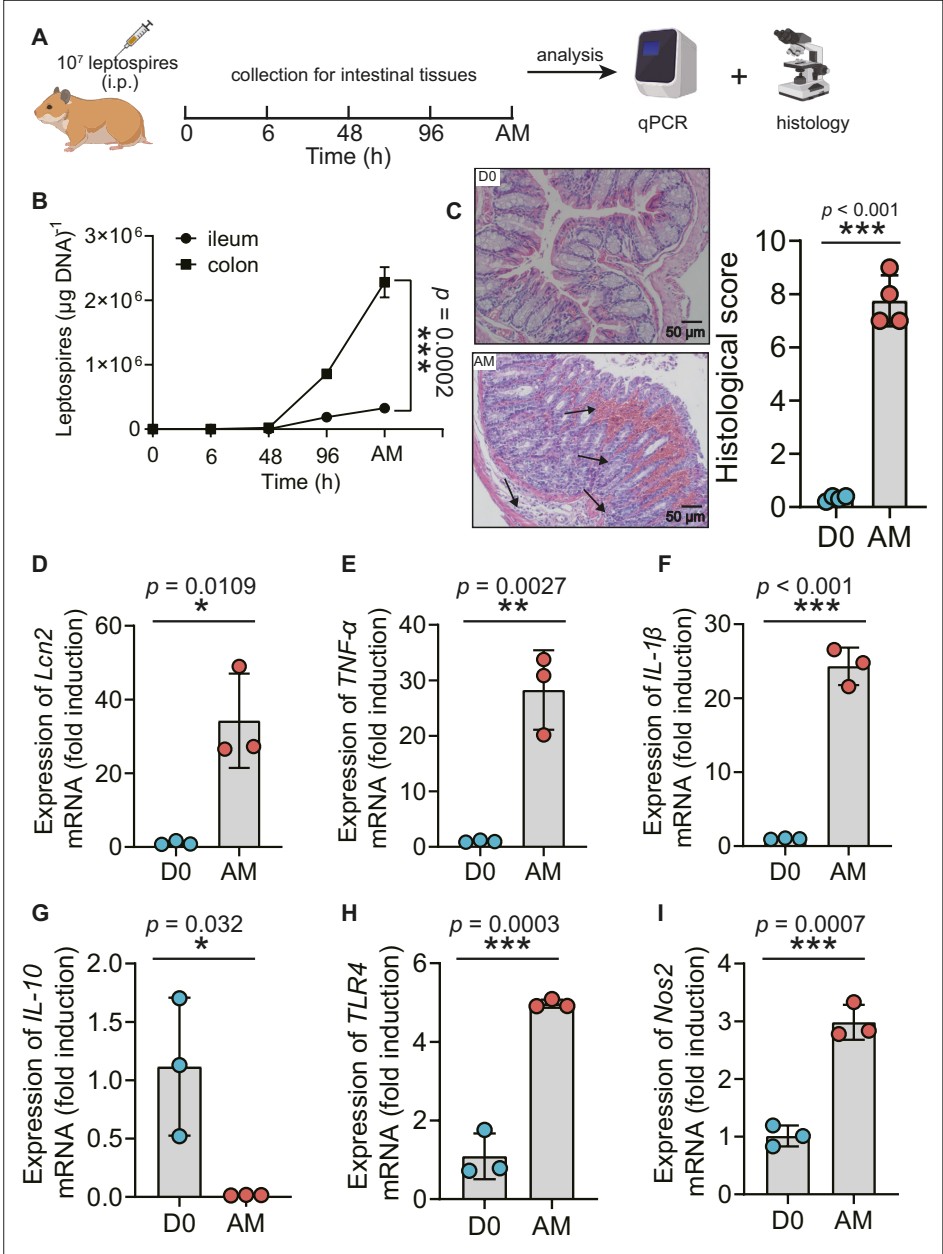

**Figure 1.** *L. interrogans* proliferated in the intestine, destroyed the intestinal structure, and increased intestinal inflammation. (**A**) Flow diagram of the experiment. Six-week-old female hamsters were injected intraperitoneally with $10^7$ leptospires. Hamsters were euthanized at 0 hr, 6 hr, 48 hr, 96 hr, and AM post infection (p.i.). The ileums and colons were collected aseptically for further analysis. (**B**) Leptospiral burdens in the ileums and colons of *Leptospira*-infected hamsters were determined by qPCR (n = 6). (**C**) Left panel: histopathology of the colons of D0 hamsters (scale bar, 50 μm; n = 4) and AM hamsters (scale bar, 50 μm; n = 4). Representative photographs are presented. Right panel: histopathology scores of the colons. D0, uninfected hamster; AM, articulo mortis. (**D–I**) The gene expression of *Lcn2* (**D**), *TNF-α* (**E**), *IL-1β* (**F**), *IL-10* (**G**), *TLR4* (**H**), and *Nos2* (**I**) was analyzed by RT-qPCR. The mRNA levels of genes in the colons of the D0 group (n = 3) and the AM group (n = 3) were normalized to the expression of the housekeeping gene *GAPDH*. Each infection experiment was repeated three times. Data are shown as the mean ± SEM and analyzed by using the Wilcoxon rank-sum test. *p<0.05, **p<0.01, ***p<0.001.

The online version of this article includes the following source data and figure supplement(s) for figure 1:

**Source data 1.** Original data for data analysis in *Figure 1B–I*.

**Figure supplement 1.** The effect of challenge routes on leptospirosis.

**Figure supplement 1—source data 1.** Original data for data analysis in *Figure 1—figure supplement 1A and B*.

*Figure 1 continued on next page*

*Figure 1 continued*

**Figure supplement 2.** The gene expression of TLRs in the intestine and inflammatory cytokines in the blood.

**Figure supplement 2—source data 1.** Original data for data analysis in *Figure 1—figure supplement 2A–H*.

other TLRs (*Aldape et al., 2010*). The results showed that the gene expression of *TLR2* and *TLR9* was significantly increased, while *TLR3, TLR5,* and *TLR7* showed no obvious difference in the AM group compared with the D0 group (*Figure 1—figure supplement 2A–E*). It is well known that *Leptospira* infection can cause leptospiremia and cytokine storms at late stages (*Cagliero et al., 2018*; *Limothai et al., 2021*). Our results also proved that the gene expression of *TNF-α, IL-1β,* and *IL-10* was increased in the blood of the AM group compared with that of the D0 group (*Figure 1—figure supplement 2F–H*). These results indicate that *L. interrogans* was able to proliferate in the intestine and impair the intestinal structure, along with inducing intestinal inflammation and systemic inflammation.

## The disruption of the gut microbiota after *L. interrogans* infection

To obtain insight into the composition of the gut microbiota during leptospirosis, we analyzed the feces of *Leptospira*-infected hamsters by 16S rRNA gene sequencing at the indicated time points (*Figure 2A*). The results showed that the bacterial richness (observed species and ace) and the diversity (Simpson index and Shannon index) of the gut microbiota were decreased at 2 d p.i., while they were increased at AM p.i. (*Figure 2B–E*). Then, we pooled all groups together to perform principal coordinate analysis (PCoA). The results showed that the composition of the gut microbiota in the AM group was significantly different from that in the D0 group (*Figure 2G*). There was a large degree of dispersion among different individuals 2 d p.i. (*Figure 2F*), and the composition of the gut microbiota in the D2 group was also different from that of the other two groups (*Figure 2—figure supplement 1A and B*). To identify bacterial taxa that were common in the D0 group or the AM group, we performed linear discriminant analysis effect size (LEfSe) analysis. The results showed that the abundance of Proteobacteria was increased in the AM group (*Figure 2I and K*), while the abundances of *Lactobacillus* and *Allobaculum* were decreased compared with those in the D0 group (*Figure 2I, L, and M*). The composition of the bacterial taxa in the D2 group was also various compared with that of the other two groups, especially the abundance of *Lactobacillus* was also decreased in the AM group compared with that of the D2 group (*Figure 2—figure supplement 1C and D*). Although the relative abundance of Firmicutes seemed to gradually decrease after *Leptospira* infection (*Figure 2H*), the ratio of Firmicutes to Bacteroidetes showed no difference in the D0 and AM groups (*Figure 2J*). These results demonstrated that *L. interrogans* infection changed the composition of the gut microbiota of hamsters.

## *L. interrogans* infection increased intestinal permeability by disrupting tight junctions

The intestinal barrier is essential for maintaining intestinal homeostasis (*Turner, 2009*). Epithelial tight junctions define the paracellular permeability of the intestinal barrier, which is related to some tight junction proteins, such as claudins, occludin, and zona occludens 1 (Zo-1) (*Horowitz et al., 2023*). The epithelium separates the organism from the external environment and defines individual compartments within tissues. At some sites, the epithelial cells form a nearly complete barrier, the disruption of which is catastrophic (*Horowitz et al., 2023*). Most studies have relied on a probe, FITC-4 kDa dextran, which has a hydrodynamic diameter of 28 Å, to probe intestinal permeability (*Tsai et al., 2017*; *Zeng et al., 2023*; *Yang et al., 2022*). As shown in *Figure 1B*, we detected leptospiral proliferation in the gut. We wondered whether *Leptospira* infection could influence intestinal permeability. The results showed that the intestinal permeability of hamsters was increased significantly after infection (*Figure 3A*). Then, we further investigated the gene expression of tight junction proteins and *Mucin-2* in hamsters. The results showed that the gene expression of *Claudin-2* was increased, while the gene expression of *Claudin-3, JAMA, ZO-1,* and *Mucin-2* was decreased in the AM group compared with the D0 group (*Figure 3B–F*).

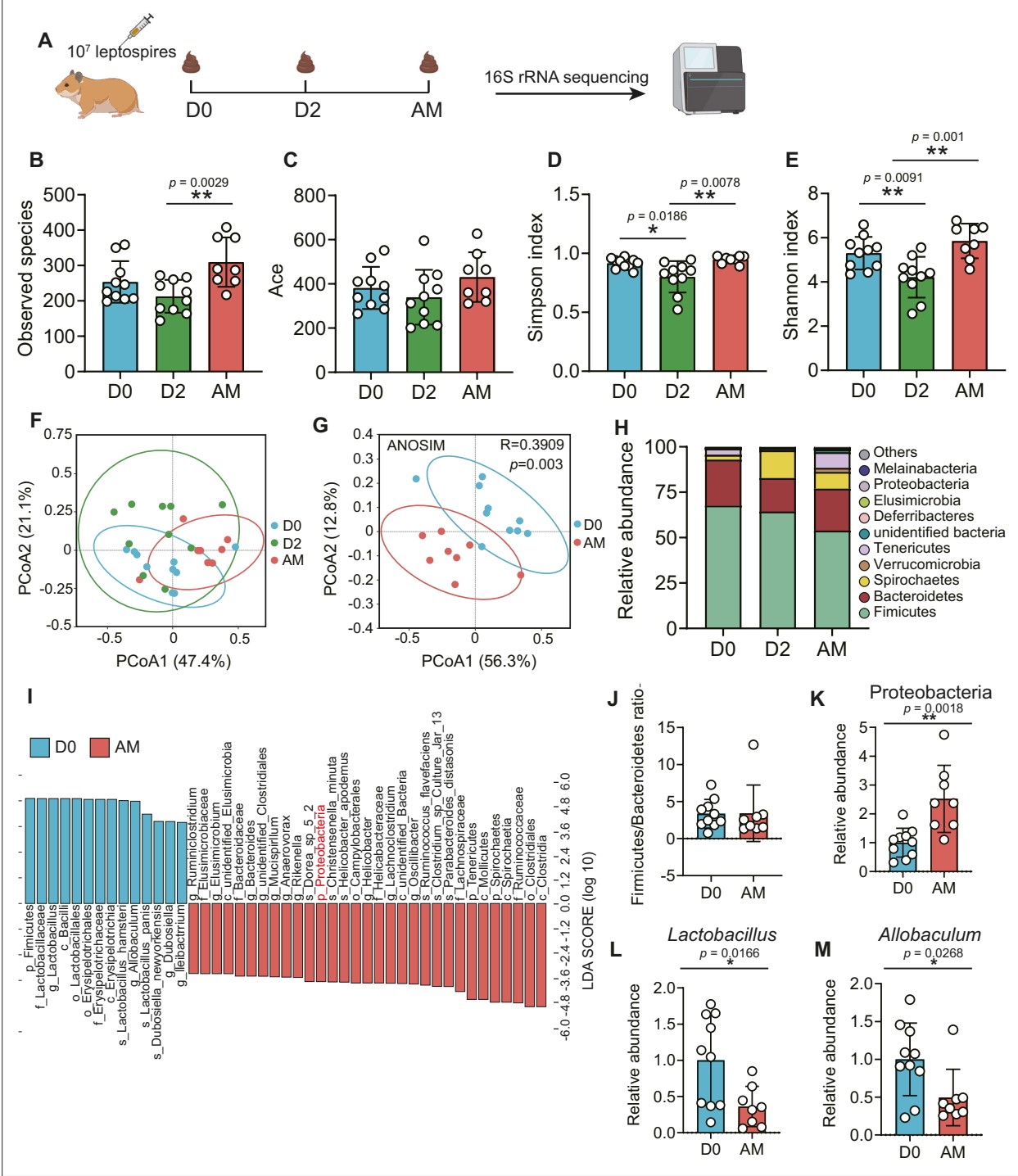

**Figure 2.** *L. interrogans* infection changed the composition of the gut microbiota. (**A**) Flow diagram of the experiment. Six-week-old female hamsters were injected intraperitoneally with $10^7$ leptospires. Fecal pellets were collected aseptically for 16S rRNA gene sequencing at 0 d, 2 d, and AM post infection (p.i.). D0, uninfected hamster; AM, articulo mortis. (**B, C**) Observed species (**B**) and ace (**C**) in the feces of hamsters. Observed species and ace indicate species richness. (**D, E**) Simpson index (**D**) and Shannon index (**E**) in the feces of hamsters. Simpson and Shannon indexes indicate species diversity. (**F**) Principal coordinate analysis (PCoA) of fecal samples based on 16S rRNA gene sequencing using weighted UniFrac. (**G**) PCoA of fecal samples based on 16S rRNA gene sequencing using weighted UniFrac. (**H**) Relative abundance of the top 10 phyla in the feces of hamster. (**I**) Linear discriminant analysis (LDA) of effect size (LEfSe) between D0 and AM hamsters (LDA score > 3). Cambridge blue bars indicate taxa enrichment in D0 hamsters, and pink bars indicate taxa enrichment in AM hamsters. (**J**) The ratio of Firmicutes to Bacteroidetes in the D0 and AM groups. (**K–M**) The relative abundance of Proteobacteria (**K**), *Lactobacillus* (**L**), and *Allobaculum* (**M**) in the D0 and AM groups (D0, n = 10; D2, n = 10; AM, n = 8). Data are shown as the mean ± SEM and analyzed by using the Wilcoxon rank-sum test. *p<0.05, **p<0.01.

*Figure 2 continued on next page*

*Figure 2 continued*

The online version of this article includes the following source data and figure supplement(s) for figure 2:

**Source data 1.** Original data for data analysis in *Figure 2B–E, J–M*.

**Figure supplement 1.** The effect of *Leptospira* infection on the composition of the gut microbiota.

### *L. interrogans* infection caused the translocation of Proteobacteria

Damaged intestinal structure may increase the translocation of pathobionts (*Muñoz et al., 2019*; *Bertocchi et al., 2021*), and our 16S rRNA sequencing showed an increase in Proteobacteria in the AM group. We hypothesized that the pathogenic Proteobacteria could translocate into the intestinal epithelium or bloodstream after infection. First, we found that the intestinal epithelium of the AM group had more bacteria than that of the D0 and D2 groups, as determined by the EUB 338 probe, which was supposed to hybridize with all bacteria (*Figure 4*). Then, we examined the intestines of the D0 group and the AM group by fluorescence in situ hybridization (FISH) assays with the GAM42a probe, which was supposed to hybridize γ-Proteobacteria, and immunofluorescence (IF) assays with the anti-*L. interrogans* serovar Lai strain Lai 56601 antiserum. The results showed that the intestinal epithelium of the AM group indeed had γ-Proteobacteria that was not colocalized with leptospires (*Figure 4—figure supplement 1A*). Then, we tested whether the gut microbiota could translocate into the blood. However, no bacteria were recovered from the blood by aerobic or anaerobic culture on LB and MacConkey agar plates (*Figure 4—figure supplement 1B*).

### Gut microbiota depletion exacerbated leptospirosis

To determine the role of the gut microbiota in hamsters during *L. interrogans* infection, we treated hamsters with broad-spectrum antibiotics to deplete the gut microbiota, as shown in *Figure 5A*. 16S rRNA sequencing analysis indicated that the richness of the gut microbiota decreased after antibiotic treatment (*Figure 5B and C*). In addition, the diversity of the gut microbiota in the Abx-treated

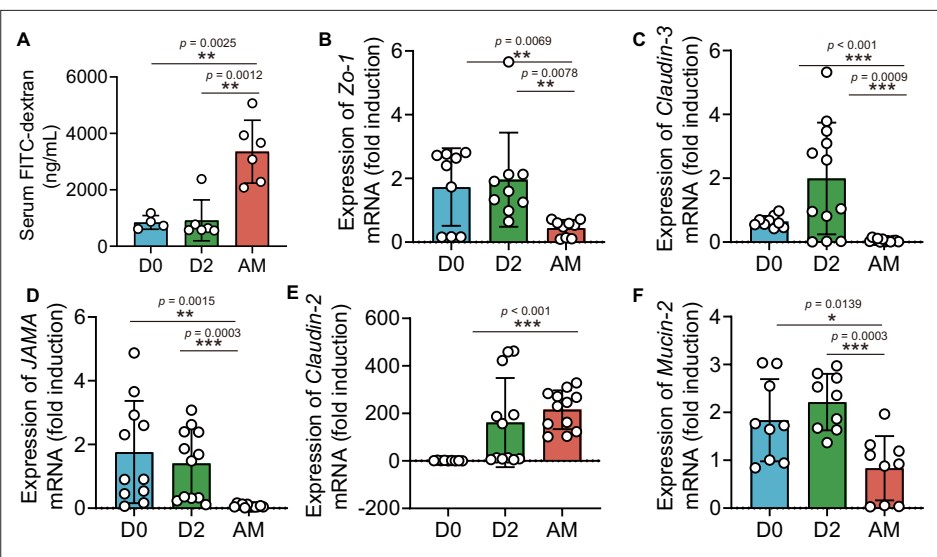

**Figure 3.** *L. interrogans* infection disrupted intestinal tight junctions. Six-week-old female hamsters were injected intraperitoneally with $10^7$ leptospires. Blood was collected to determine intestinal permeability at 0 d, 2 d, and AM post infection (p.i.). Colons were collected aseptically to measure gene expression at 0, 2, and AM p.i. (**A**) The intestinal permeability of *Leptospira*-infected hamsters was analyzed with a fluorescence spectrophotometer at the indicated time (D0, n = 4; D2, n = 6; AM, n = 6). (**B–F**) The relative gene expression of *ZO-1* (**B**), *Claudin-3* (**C**), *JAMA* (**D**), *Claudin-2* (**E**), and *Mucin-2* (**F**) in *Leptospira*-infected hamsters (n = 6–10 per group) was determined by qPCR. D0, uninfected hamster. AM, articulo mortis. Each infection experiment was repeated three times. Data are shown as the mean ± SEM and analyzed by using the Wilcoxon rank-sum test. *p<0.05, **p<0.01, ***p<0.001.

The online version of this article includes the following source data for figure 3:

**Source data 1.** Original data for data analysis in *Figure 3A–F*.

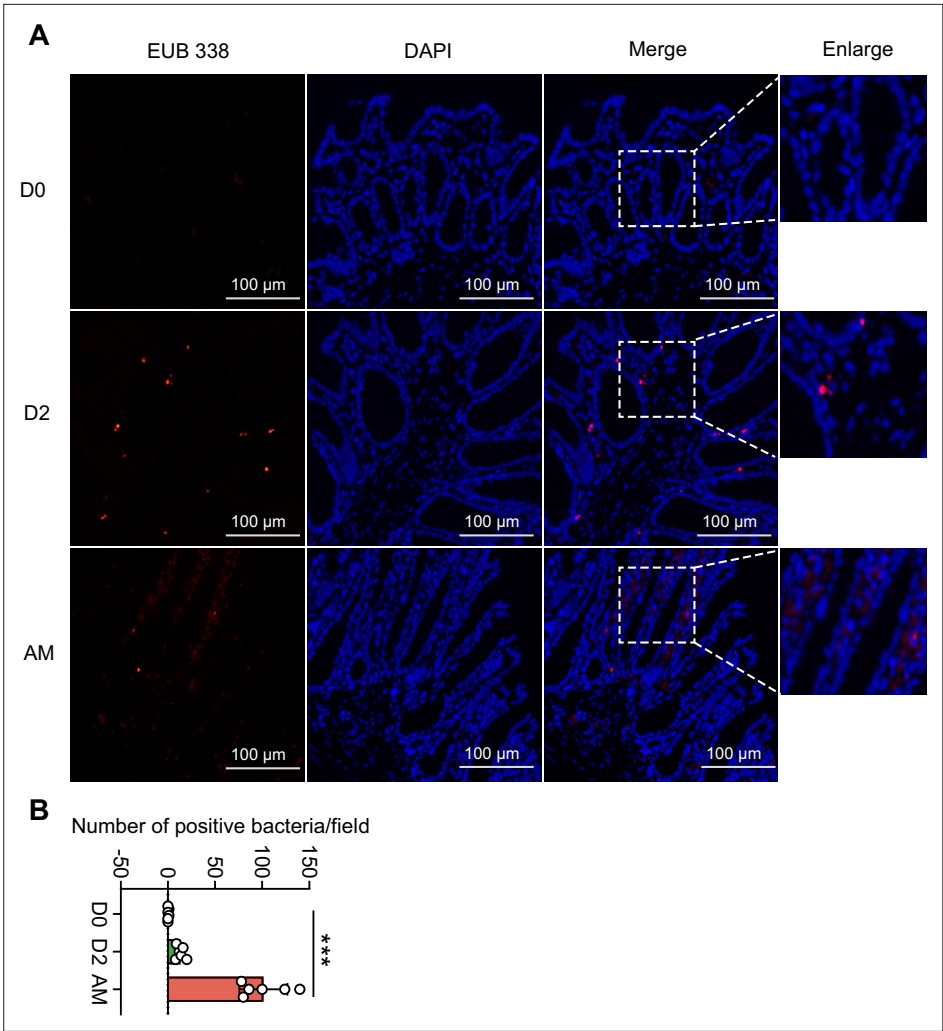

**Figure 4.** Translocated microbiota in the intestinal epithelium following *L. interrogans* infection. Six-week-old female hamsters were injected intraperitoneally with $10^7$ leptospires. Hamsters were euthanized at 0 d, 2 d, and AM post infection (p.i.). Colons were collected aseptically. The intestinal contents were excluded from the intestine with phosphate-buffered saline (PBS) and a segment of the colon was fixed in 4% paraformaldehyde solution overnight and analyzed by fluorescence in situ hybridization (FISH) with an EUB 338 probe (red). (**A**) The results are representative photographs of three groups. Scale bar, 100 μm. (**B**) Number of positive bacteria per field in the three groups. n = 6 per group. D0, uninfected hamster; AM, articulo mortis. Each infection experiment was repeated three times. Data are shown as the mean ± SEM and analyzed by using the Wilcoxon rank-sum test. ***p<0.001.

The online version of this article includes the following source data and figure supplement(s) for figure 4:

**Source data 1.** Original data for data analysis in *Figure 4B*.

**Figure supplement 1.** Translocation of the Proteobacteria during *L. interrogans* infection.

hamsters was lower than that of the control group, and this change in diversity was reversed by fecal microbiota transplantation (FMT) (*Figure 5D and E*). Antibiotic treatment changed the composition of the gut microbiota in hamsters (*Figure 5F*, *Figure 5—figure supplement 1A–D*), leading to the expansion of Proteobacteria and a decrease in Bacteroidota (*Figure 5F and G*). The survival time of Abx-treated hamsters was significantly reduced after infection with $10^6$ leptospires (*Figure 5I*). In addition, we also measured leptospiral load and the gene expression of inflammatory cytokines in the blood. The results showed that the leptospiral load of the Abx group was significantly increased compared with that of the control group, and this increase was partially reversed by FMT (*Figure 5J*). Abx treatment slightly increased the gene expression of *TNF-α* and *IL-1β* compared with those of

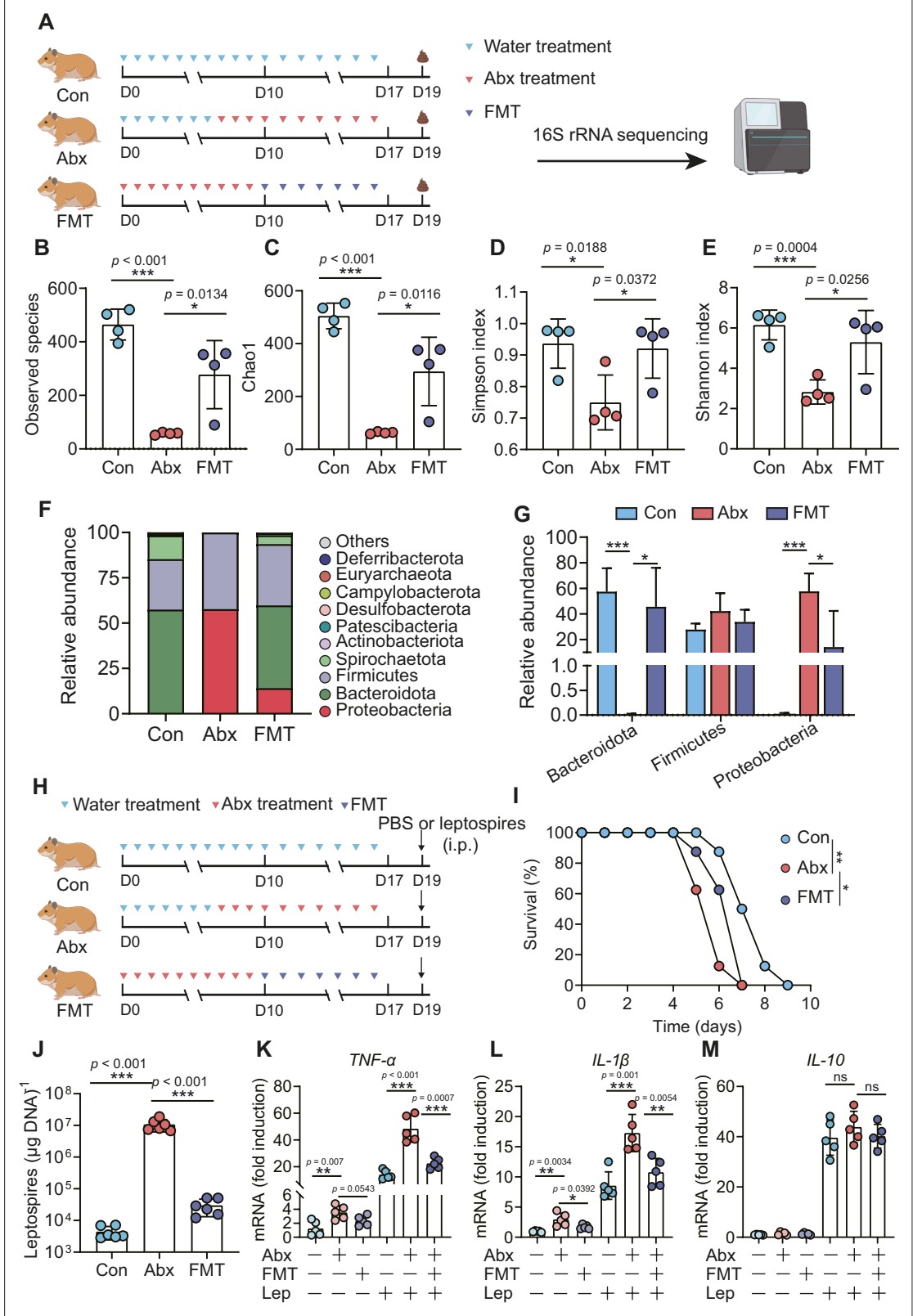

**Figure 5.** Microbiota depletion exacerbated leptospirosis. (**A**) Flow diagram of the experiment. Hamsters were administered with Abx or water intragastrically once daily for 10 consecutive days. Then, fecal pellets were collected aseptically for 16S rRNA gene sequencing. (**B, C**) Observed species (**B**) and Chao1 (**C**) in the feces of hamsters. Observed species and Chao1 indicate species richness. (**D, E**) Simpson index (**D**) and Shannon index (**E**) in the feces of hamsters. Simpson and Shannon indexes indicate species diversity. (**F**) Relative abundance of the top 10 phyla in the feces of hamsters.

*Figure 5 continued on next page*

*Figure 5 continued*

(**G**) The relative abundance of Bacteroidetes, Firmicutes, and Proteobacteria in the Con, Abx, and fecal microbiota transplantation (FMT) groups. n = 4 per group. (**H**) Hamsters were treated with Abx as described above. After a 2-day washout period, hamsters were intraperitoneally infected with 10⁶ *L. interrogans*. Then, the survival rate of the hamsters was recorded. n = 6 per group. (**J**) Leptospiral load in the blood of the Con, Abx, and FMT groups. n = 6 per group. (**K, M**) Gene expression of *TNF-α* (**K**), *IL-1β* (**L**), and *IL-10* (**M**) in the blood of the three groups. n = 5 per group. Each infection experiment was repeated three times. Data are shown as the mean ± SEM and analyzed by using the Wilcoxon rank-sum test. Survival differences between the study groups were compared by using the Kaplan–Meier log-rank test. *p<0.05, **p<0.01, ***P<0.001. ns, not significant.

The online version of this article includes the following source data and figure supplement(s) for figure 5:

**Source data 1.** Original data for data analysis in *Figure 5B–D, E, G, I–M*.

**Figure supplement 1.** The effect of Abx treatment on the composition of the gut microbiota.

the untreated group. Consistent with this result, the Abx-treated infection group exhibited higher expression of *TNF-α* and *IL-1β* than the untreated infection group, which was also partially reversed by FMT (*Figure 5K and L*). Although *Leptospira* infection significantly increased the gene expression of *IL-10* compared with that of the untreated group, there was no significant difference among the control infection group, the Abx-treated infection group, and the FMT infection group (*Figure 5M*). These results indicated that gut microbiota homeostasis helped fight *Leptospira* infection in hamsters.

## Gut-derived LPS in the intestine and serum-induced inflammation

Gut dysbiosis is a major determinant of endotoxemia via dysfunction of the intestinal barrier scaffold, which is a prerequisite for LPS translocation into systemic circulation (*Violi et al., 2023*). A previous study demonstrated that gut-derived LPS was a central mediator in alcoholic steatohepatitis (*Szabo, 2015*), white adipose tissue inflammation (*Caesar et al., 2012*), and peripheral inflammation (*Lappi et al., 2013*). Cytokine storms are characteristic of severe leptospirosis (*Cagliero et al., 2018*). Our results showed that the proportion of Proteobacteria and intestinal permeability was increased in the AM group; thus, we hypothesized that gut-derived LPS promoted the inflammatory response in severe leptospirosis. We found that the LPS level was higher in the serum of the AM group than that of the D0 group (*Figure 6—figure supplement 1B*). The LPS level in the Abx-treated AM group was lower than that in the phosphate-buffered saline (PBS)-treated AM group (*Figure 6—figure supplement 1B*). To further gain insight into the role of gut-derived LPS in inflammation, we cocultured macrophages with different concentrations of fecal filtration or serum and measured the changes in gene expression (*Figure 6A*). The results showed that fecal filtration from uninfected hamsters and AM hamsters both upregulated the gene expression of *IL-1β*, *IL-10*, and *TLR4*, while only fecal filtration from AM hamsters upregulated the gene expression of TNF-α (*Figure 6B–E*). The serum of uninfected hamsters significantly increased the transcription of *TNF-α* (*Figure 6F*), while the serum of AM hamsters only slightly upregulated the gene expression of *TNF-α* (*Figure 6F*). Serum from uninfected hamsters and AM hamsters promoted the transcription of IL-1β (*Figure 6G*). However, only the serum of AM hamsters significantly upregulated the gene expression of *IL-10* and *TLR4* (*Figure 6H and I*). The identification of TLR4, a well-known receptor of LPS, prompted us to examine whether the proinflammatory activity of fecal filtration and serum of AM hamsters was mediated by LPS. A previous study demonstrated that cytokine responses to heat-killed leptospires were not inhibited by polymyxin B (*Viriyakosol et al., 2006*). Our results showed that LPS neutralization with PMB inhibited gene expression of *TNF-α*, *IL-1β*, *IL-10*, and *TLR4* in macrophages after stimulation by fecal filtration or serum from AM hamsters (*Figure 6J–Q*). These results indicated that gut-derived LPS might be an essential factor that causes inflammatory storm in hamsters with severe leptospirosis.

## LPS neutralization synergized with antibiotic therapy or polyclonal antibody therapy improved the survival rate in severe leptospirosis

Based on the above results, we proposed that in addition to leptospires, gut-derived LPS might also play a pathogenic role in severe leptospirosis and induce an uncontrolled inflammatory response, in which case neutralizing LPS could be effective in treating severe leptospirosis. Therefore, we explored the effect of LPS neutralization therapy compared with antibiotic therapy and our published rabbit polyclonal antibody therapy in severe leptospirosis (*Jin et al., 2016*). As shown in *Figure 7A*, moribund hamsters were treated twice a day for three consecutive days. LPS neutralization therapy,

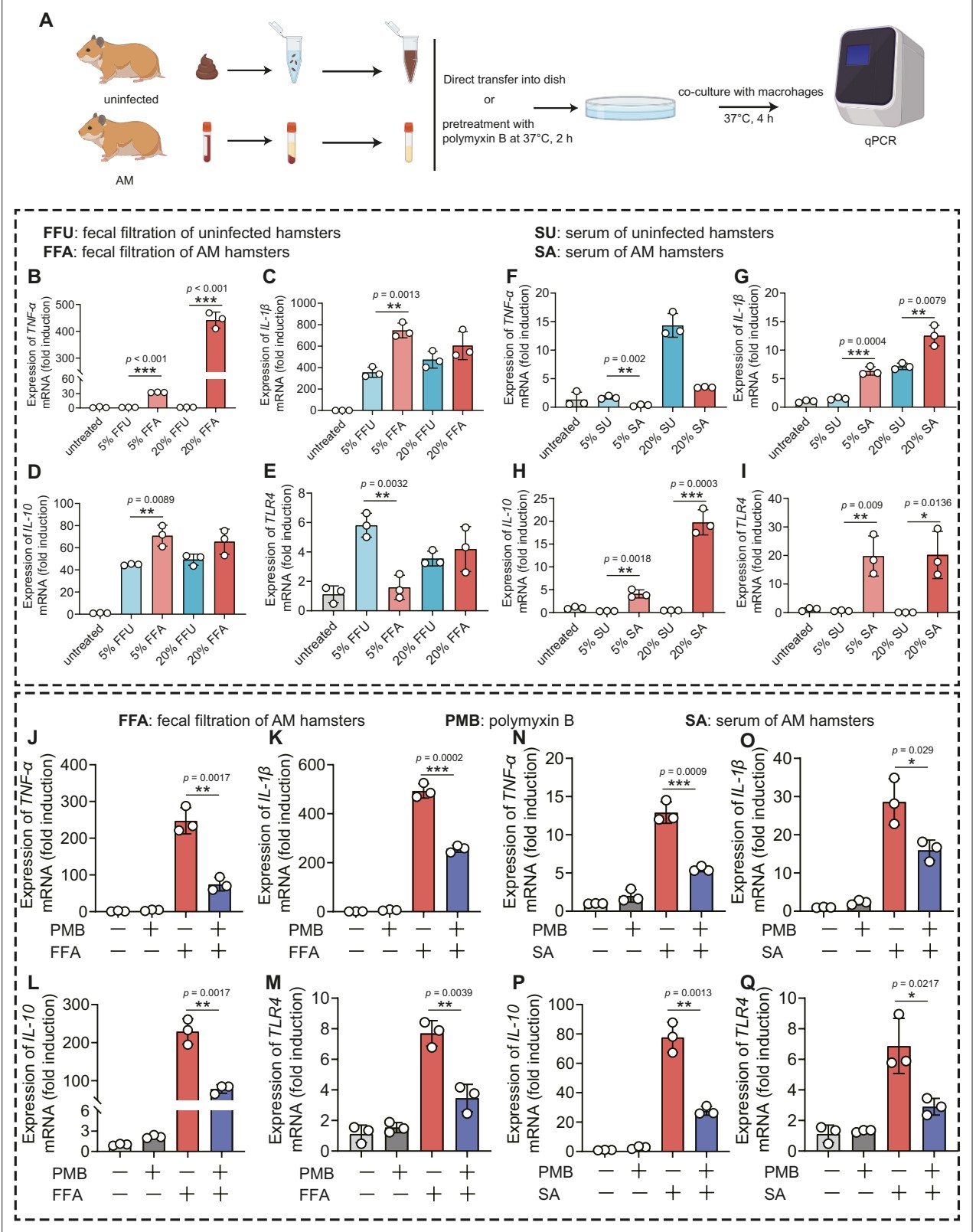

**Figure 6.** *Leptosira* infection induced intestinal and systemic inflammation that was partially inhibited by lipopolysaccharide (LPS) neutralization. (**A**) Flow diagram of the experiment. Fecal filtration or serum of uninfected hamsters and AM hamsters was directly added into dishes or pretreated with polymyxin B (PMB) at 37°C for 2 hr, and then cocultured with macrophages for 4 hr. RNA was extracted for gene expression analysis. (**B–E**) The gene expression of *TNF-α* (**B**), *IL-1β* (**C**), *IL-10* (**D**), and *TLR4* (**E**) of fecal filtration of uninfected hamsters (n = 3) and AM hamsters (n = 3) was analyzed by

*Figure 6 continued on next page*

*Figure 6 continued*

RT-qPCR. (**F–I**) The gene expression of *TNF-α* (**F**), *IL-1β* (**G**), *IL-10* (**H**), and *TLR4* (**I**) of serum of uninfected hamsters (n = 3) and AM hamsters (n = 3) was analyzed by RT-qPCR. (**J–M**) The effect of LPS neutralization on the gene expression of *TNF-α* (**J**), *IL-1β* (**K**), *IL-10* (**L**), and *TLR4* (**M**) in the fecal filtration of AM hamsters (n = 3) was analyzed by RT-qPCR. (**N–Q**) The effect of LPS neutralization on the gene expression of *TNF-α* (**N**), *IL-1β* (**O**), *IL-10* (**P**), and *TLR4* (**Q**) in the serum of AM hamsters (n = 3) was analyzed by RT-qPCR. The mRNA levels in untreated controls were set as 1.0. Each infection experiment was repeated three times. Data are shown as the mean ± SEM and analyzed by using the Wilcoxon rank-sum test. *$p < 0.05$, **$p < 0.01$, ***$p < 0.001$.

The online version of this article includes the following source data and figure supplement(s) for figure 6:

**Source data 1.** Original data for data analysis in *Figure 6B–Q*.

**Figure supplement 1.** *Leptospira* infection increased lipopolysaccharide (LPS) level in the blood.

**Figure supplement 1—source data 1.** Original data for data analysis in *Figure 6—figure supplement 1B*.

**Figure supplement 2.** Determination minimum inhibitory concentrations of polymyxin B (PMB) and Dox on *Leptospira* 56601.

**Figure supplement 2—source data 1.** Original data for data analysis in *Figure 6—figure supplement 2B*.

antibiotic therapy, and rabbit polyclonal antibody therapy did not significantly improve the conditions of female hamsters with severe leptospirosis. However, LPS neutralization combined with antibiotic therapy or rabbit polyclonal antibody therapy significantly increased the survival rate up to 50 or 66.7% (*Figure 7B*). To further explore the factors leading to differences in survival, we determined the leptospiral load and gene expression of inflammatory cytokines in the blood at 6 d p.i. The results showed that antibiotic therapy and rabbit polyclonal antibody therapy decreased the leptospiral load in the blood, while LPS neutralization therapy did not reduce the leptospiral load in the blood (*Figure 7C*). Although antibiotic therapy reduced the leptospiral load, it significantly increased the gene expression of proinflammatory cytokines (*TNF-α* and *IL-1β*) compared with that of the control group (*Figure 7D and E*). LPS neutralization therapy combined with *Leptospira*-killing therapy decreased the gene expression of inflammatory cytokines (*TNF-α*, *IL-1β*, and *IL-10*) compared with that of the control group (*Figure 7D–F*). In addition, we examined the effectiveness of LPS neutralization therapy in male hamsters with severe leptospirosis and in female hamsters with different serotype. The results showed that LPS neutralization synergized with antibiotic therapy or polyclonal antibody therapy improved the conditions of male hamsters with severe leptospirosis and female hamsters infected by another pathogenic leptospiral serotype (*Figure 7—figure supplements 1B and 2B*). Our results indicated that LPS neutralization could be a novel therapy for severe leptospirosis.

## Discussion

Recently, Inamasu et al. investigated the pathological changes and distribution of leptospires in the gut of hamsters after subcutaneous infection (*Inamasu et al., 2022*). Their study highlighted that not only the urine of infected animals but also the feces could be a source of infection. However, the exact role of the gut microenvironment in the pathogenesis of leptospirosis is still unknown. In our study, we found that leptospires could proliferate in the ileum and colon following intraperitoneal challenge or subcutaneous challenge, which was consistent with the results of Inamasu's study (*Inamasu et al., 2022*). Hamsters infected intraperitoneally exhibited more rapid death than hamsters infected subcutaneously. A previous study reported that the route of infection affected the kinetics of hematogenous dissemination, kidney colonization, and inflammation (*Maruoka et al., 2021*). In addition, the subcutaneous route induced a slower-progressing infection than the intraperitoneal route in a leptospirosis rat model (*Zilber et al., 2016*). However, there was a higher leptospiral load in the colons than in the ileums of hamsters infected by the intraperitoneal route. The capacity of *L. interrogans* to breach the immune defense at each port of entry is determined when *Leptospira* gained access to the blood. Although intraperitoneal challenge may be not a natural route of infection, both intraperitoneal challenge and subcutaneous challenge led to the colonization of leptospires in the intestine, indicating that leptospiral colonization in the intestine was independent of the routine of infection. *Leptospira* infection destroyed the intestinal structure and induced intestinal inflammation in the colon. However, Inamasu et al. reported that *Leptospira* was not involved in the pathogenesis of the colon, suggesting that the jejunum and ileum are the main sites of lesions. Differences in bacterial strains and the route of infection might lead to variations in results. Infection via the intraperitoneal and subcutaneous routes bypasses the epidermal defense system. Early damage to the intestinal structure and environment

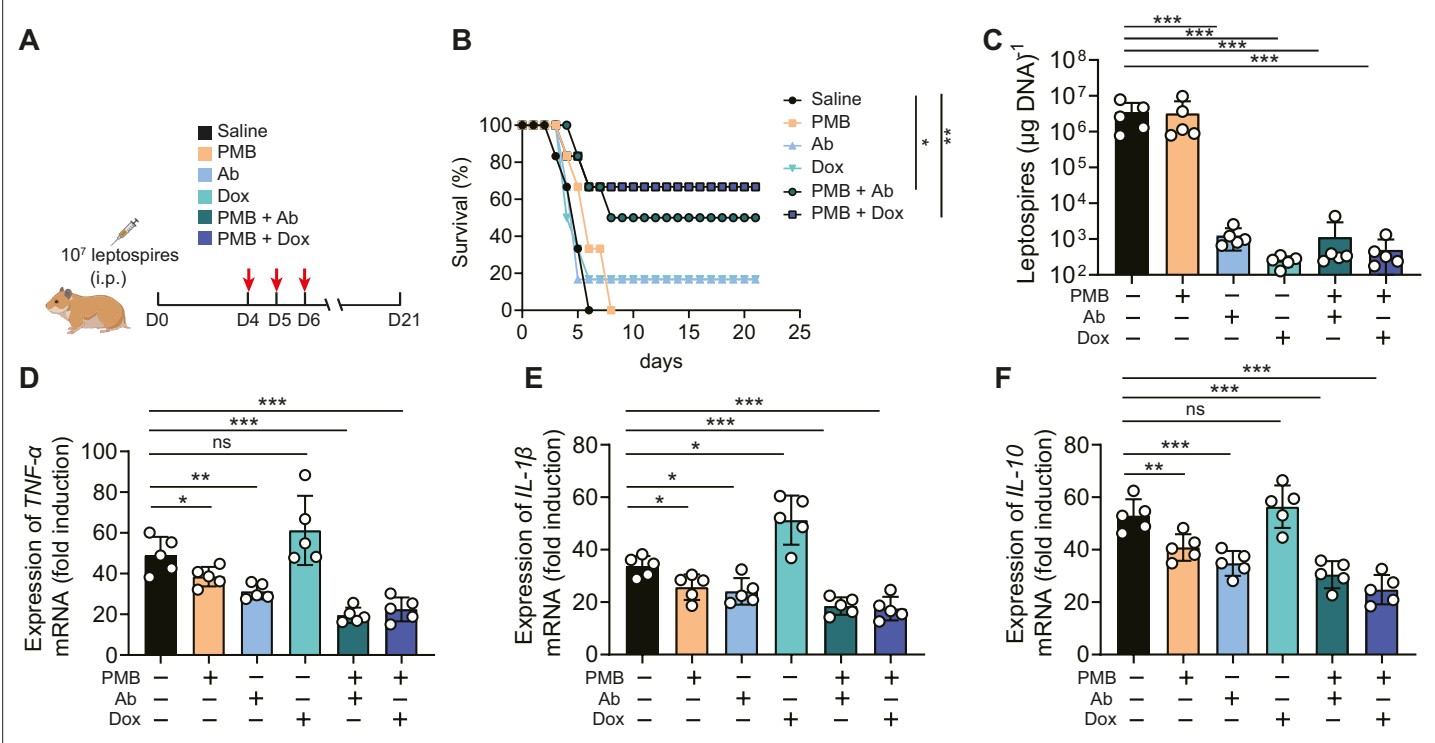

**Figure 7.** Lipopolysaccharide (LPS) neutralization combined with antibody therapy or antibiotic therapy improved the survival rate of female hamsters with severe leptospirosis. (**A**) Flow diagram of the experiment. Six-week-old female hamsters were injected intraperitoneally with $10^7$ leptospires. Group 1: saline control; group 2: polymyxin B (PMB) (1 mg/kg, i.p.); group 3: antibody against *Leptospira* (Ab) (16 mg/kg, subcutaneous injection); group 4: doxycycline (Dox) (5 mg/kg, i.p.); group 5: PMB (1 mg/kg, i.p.) and Ab (16 mg/kg, subcutaneous injection); group 6: PMB (1 mg/kg, i.p.) and Dox (5 mg/kg, i.p.). Hamsters were treated twice a day for three consecutive days. Hamsters were monitored daily for 21 d. (**B**) Survival rate of female hamsters of severe leptospirosis after different treatments. n = 6 per group. (**C**) Leptospiral load in the blood of different groups. n = 5 per group. (**D–F**) Gene expression of *TNF-α* (**D**), *IL-1β* (**E**), and *IL-10* (**F**) in the blood of different groups. The mRNA levels of cytokines in uninfected controls were set as 1.0. n = 5 per group. Each infection experiment was repeated three times. Survival differences between the study groups were compared by using the Kaplan–Meier log-rank test. *p<0.05, **p<0.01, ***p<0.001. ns, not significant.

The online version of this article includes the following source data and figure supplement(s) for figure 7:

**Source data 1.** Original data for data analysis in *Figure 7B–F*.

**Figure supplement 1.** Lipopolysaccharide (LPS) neutralization combined with antibody therapy or antibiotic therapy improved the survival rate in male hamsters of severe leptospirosis.

**Figure supplement 1—source data 1.** Original data for data analysis in *Figure 7—figure supplement 1B*.

**Figure supplement 2.** Lipopolysaccharide (LPS) neutralization combined with antibiotic therapy improved the survival rate in female hamsters infected by 56606.

**Figure supplement 2—source data 1.** Original data for data analysis in *Figure 7—figure supplement 2B*.

may be one of the causes of rapid death following infection by intraperitoneal injection. The proliferation of leptospires in the intestine makes feces a potential excretion route. Our previous study demonstrated that leptospires were rapidly cleared in the intestine and that *Leptospira* infection did not increase the intestinal permeability of mice (*Xie et al., 2022*). A recent study reported that pathogenic *Leptospira* spp. were highly present in human and gorilla stool samples in the Congo but were absent in other African nonhuman primates, thereby suggesting a possible gorilla–humans transfer (*Medkour et al., 2021*). Our 16S rRNA sequencing results also showed that the relative abundance of Spirochetes was increased in the AM group compared with that of the D0 group, which might be related to the excretion of leptospires in the feces. However, no leptospires were detected in the feces of *Leptospira*-infected rats (*Zilber et al., 2016*). Serovar specificity and species differences may determine the excretion route of leptospires and the disruption of intestinal homeostasis may influence the progression of leptospirosis.

Our 16S rRNA sequencing results showed that the species richness and diversity of the gut microbiota in the AM group were even higher than those in the D2 group. The species richness and diversity of the gut microbiota decreased first and then increased with disease progression. This phenomenon was also reported in melioidosis and HIV-1 infection (*Lankelma et al., 2017*; *Lozupone et al., 2013*). Lankelma et al. explained that several 'big players' in the gut microbiota were eliminated by the host immune system during severe systemic bacterial infection, giving way to the proliferation of other bacteria (*Lankelma et al., 2017*). Specific microbiota community composition, rather than species richness, drives microbiota-mediated resistance against *Leptospira* infection. Becattini et al. demonstrated that microbiota composition and the type of immune stimulus impacted bacterial responses and that the intestinal metabolome was rapidly reshaped by host immune activation (*Becattini et al., 2021*). We hypothesized that the early colonization and proliferation of leptospires in the intestine rapidly activated intestinal immunity, leading to the expansion of specific microbes in the AM group. The composition of the gut microbiota in the D2 group was discrete, as shown by PCoA. This difference might be related to the disease progression. In addition, we found that the levels of *Lactobacillus* and *Allobaculum* were higher in the D0 group than that in the AM group at the genus level. Meanwhile, the level of Proteobacteria increased in the AM group. A previous study demonstrated that pretreatment with *Lactobacillus plantarum* prevented severe pathogenesis in mice (*Potula et al., 2017*), which highlighted the *Lactobacillus* as candidate for leptospirosis prevention. Our study proved that *Leptospira* infection increased the gene expression of *TLR2*, *TLR4*, *TLR9*, and some proinflammatory cytokines, which created a proinflammatory environment in the intestine. A previous study demonstrated that intestinal inflammation increased the bioavailability of respiratory electron acceptors for Enterobacteriaceae, which promoted the expansion of pathogenic Enterobacteriaceae (*Byndloss et al., 2017*). Therefore, it is plausible that *Leptospira* infection triggered intestinal inflammation, which promoted the expansion of Proteobacteria by increasing the bioavailability of respiratory electron acceptors for pathogenic Proteobacteria.

The intestinal barrier is critical for maintaining homeostasis in the host. The apical junctional complex (AJC) encircles pairs of neighboring epithelial and endothelial cells to create an adhesive network and maintain barrier integrity (*Rusu and Georgiou, 2020*). Our results showed that *Leptospira* infection significantly disrupted the intestinal permeability of hamsters at the late stage of infection. Previous studies have demonstrated that *Leptospira* infection reduced the signaling between cell–cell junction proteins and caused the mislocalization of AJC proteins, such as occludin and Zo-1 (*Sebastián et al., 2021*). A recent study reported that *L. interrogans* could disassemble AJCs in renal proximal tubule epithelial cells and transmigrate through the paracellular route for dissemination in the host (*Sebastián et al., 2021*). Thus, we hypothesize that *L. interrogans* was also able to transmigrate through the intestinal epithelial barrier, and feces excretion route of leptospires was possible in susceptible hosts during leptospirosis. Our results showed that the gene expression of *Claudin-2*, *Claudin-3*, *Claudin-4*, *Zo-1*, and *Mucin-2* was compensatively upregulated in the D2 group compared with those in the D0 group. However, the gene expression of *Claudin-3*, *JAMA*, *Zo-1*, and *Mucin-2* was downregulated in the AM group compared with that in the D0 group. In contrast, the gene expression of *Claudin-2* was higher in the AM group than that in the D0 group, which might explain why moribund hamsters had diarrhea (*Tsai et al., 2017*).

Gut microbiota-depleted hamsters exhibited reduced survival time, increased leptospiral load, and upregulated gene expression of proinflammatory cytokines compared with untreated hamsters after *Leptospira* infection, while FMT partially reversed these effects. Our previous study demonstrated that the leptospiral load was increased in the organs of gut microbiota-depleted mice and that gut microbiota depletion diminished the bactericidal activity of macrophages during *Leptospira* infection (*Xie et al., 2022*). These results indicated that the homeostasis of the gut microbiota is essential for both susceptible and resistant hosts to combat *Leptospira* infection. 16S rRNA sequencing analysis revealed that antibiotic treatment decreased the diversity of the gut microbiota, and FMT partially reversed these changes. Antibiotic treatment increased the relative abundance of proinflammatory bacteria, such as γ-Proteobacteria, while it reduced the relative abundance of anti-inflammatory bacteria, such as *Allobaculum*. The alteration of the gut microbiota might skew the host to an inflammatory condition and reduce antileptospiral capacity. FMT aims to replace an unfavorable resident gut microbiota with a favorable microbiota from a healthy donor. FMT has demonstrated clear efficacy in the treatment of recurrent *Clostridioides difficile* infection (rCDI), inflammatory bowel disease, and

neurological disorders (*Schmidt et al., 2022*). The efficiency of FMT determines the phenotypes of subjects in the infection model. Prolonging the duration of FMT might help recover the gut microbiota homeostasis and the capacity to combat anti-*Leptospira* infection.

Our study showed the translocation of Proteobacteria to the intestinal epithelium. Although no bacteria were found in the blood of the AM group, the level of LPS was increased in the PBS-treated AM group compared with that of the uninfected group, and it was decreased in the Abx-treated AM group compared with that of the PBS-treated AM group. This result indicated that gut-derived LPS was involved in severe leptospirosis. Gut dysbiosis is a primary determinant of low-grade endotoxemia mediated by dysfunction of the intestinal barrier scaffold, which is a prerequisite for LPS translocation into the systemic circulation (*Violi et al., 2023*). Importantly, mice (resistant to leptospirosis) can tolerate levels of LPS endotoxin 250 higher than the level that humans can tolerate (*Copeland et al., 2005*). In addition, classical Gram-negative LPS had a 1000-fold higher potency than leptospiral LPS (*Werts et al., 2001*). Fecal filtration from the AM group and the uninfected group both increased the gene expression of *TLR4*, while only serum from the AM group increased the gene expression of *TLR4*. TLR4 expression was inhibited by PMB treatment. In severe leptospirosis, excessive immune activation is a hallmark of disease (*Cagliero et al., 2018*). A previous study demonstrated that the cytokine response to heat-killed whole leptospires was not inhibited by PMB in macrophages (*Viriyakosol et al., 2006*). Leptospiral LPS has been described as atypical in terms of structure, and silver-stained electrophoretic profiles of LPS from pathogenic leptospiral strains were shown to be different from those of *Escherichia coli* LPS (*Bonhomme et al., 2020*). Although it is still difficult to distinguish the specific roles of gut-derived LPS and leptospiral LPS in promoting inflammation, we hypothesize that gut-derived LPS may also be an essential driver of inflammatory reactions in severe leptospirosis. In a rat cirrhosis model, gut decontamination with a 2-week course of antibiotics led to a redistributed microbiota, reduced proinflammatory activation of mucosal immune cells, and diminished gut bacterial translocation (*Muñoz et al., 2019*). Another study showed that endogenous endotoxins caused a pan enteric inflammatory ileus after colonic surgery and that gut decontamination alleviated small intestinal muscularis inflammation (*Türler et al., 2007*). Our results showed that doxycycline treatment significantly reduced the leptospiral load in the blood in severe leptospirosis; however, the gene expression of proinflammatory cytokines was upregulated, which was consistent with the Jarisch–Herxheimer Reaction (*Guerrier and D'Ortenzio, 2013*). Recently, Li et al. proposed an all-in-one drug delivery strategy equipped with bactericidal, LPS neutralization, and detoxification activities to treat sepsis-associated infections and hyperinflammation (*Li et al., 2023*). Our study also highlighted the important role of endogenous gut-derived LPS induced by *Leptospira* infection. Thus, the neutralization of gut-derived LPS and antileptospiral therapy are both essential in the treatment of severe leptospirosis; thus, these strategies provide a novel avenue for the treatment of human acute leptospirosis in the future.

Our study has potentially important clinical implications for the care of critically ill patients with severe leptospirosis. In human infections, gastrointestinal symptoms such as abdominal pain, vomiting, and diarrhea have been frequently observed, which has been linked to an increased risk of organ dysfunction, and even death (*Haake and Levett, 2015*; *Jiménez et al., 2018*). Our findings implicate impairment of the intestinal barrier by *Leptospira* infection as a mechanism underlying the association between gut dysbiosis and disseminated gut-derived LPS in the ICU. More importantly, from a translational perspective, our findings suggest that the increased risk of death in severe leptospirosis can be reduced by LPS neutralization combined with anti-*Leptospira* therapy. In fact, gut microbiota dysbiosis in COVID-19 patients or HIV infection is also associated with microbial translocation, LPS biosynthesis, and an expansion of Proteobacteria (*Bernard-Raichon et al., 2022*; *Zhang et al., 2022*). The application of LPS antagonist therapies, whether microbial or pharmaceutical, may be a viable precision medicine strategy to increase the survival rate of patients with severe leptospirosis or other gut-damaging infections by protecting against the spread of LPS in the bloodstream.

## Materials and methods
### Ethics statement
A previous study reported that sex influenced host susceptibility to leptospirosis (*Gomes et al., 2018*) and that sex may affect the composition of the gut microbiota (*Vemuri et al., 2019*). To reduce

variability in infection outcomes (*Tawk et al., 2023*), most infection experiments were performed in female Syrian golden hamsters (*Mesocricetus auratus*). For experiments utilizing neutralizing LPS therapy combined with antibiotic or antibody therapy for severe leptospirosis, both sexes were used. Specific pathogen-free female Syrian hamsters were maintained on standard rodent chow with water supplied ad libitum during the experimental period. All animal experiments were conducted according to the regulations of the Administration of Affairs Concerning Experimental Animals in China. The protocol was approved by the Institutional Animal Care and Use Committee of Jilin University (20170318).

## Bacterial strains and animals

Pathogenic *L. interrogans* serovar Lai strain Lai (56601) and *L. interrogans* serovar Autumnalis (56606) were grown in liquid Ellinghausen–McCullough–Johnson–Harris (EMJH) medium at 29°C without agitation. A Petroff–Hauser chamber was used to count leptospires. Six-week-old hamsters were provided by Liaoning Changsheng Biotechnology Co. Ltd.

## Experimental infections

### Part I: Hamster infection

To explore the effect of the challenge route on leptospiral colonization in the intestine and the survival rate and whether it was specific to the routine of infection, 6-week-old female hamsters were injected intraperitoneally or subcutaneously with $10^7$ leptospires. Hamsters were euthanized at 0 hr, 6 hr, 48 hr, and 96 hr p.i., and articulo mortis stage (AM). The ileums and colons were collected aseptically. The intestinal contents were excluded from the intestine with PBS, and the tissues were stored at –80°C for leptospiral load analysis. For the survival assay, hamsters were monitored daily for signs of illness including ruffled fur, gait difficulty, loss of appetite, or weight loss of ≥10% of the animal's maximum weight, which was considered moribund (*Coutinho et al., 2014*). Hamsters were euthanized when they were moribund.

To explore the effect of *Leptospira* infection on the gut microbiota, 6-week-old female hamsters were injected intraperitoneally with a lethal dose of $10^7$ leptospires. Fresh fecal pellets were collected aseptically at 0 d, 2 d, and AM p.i., immediately frozen in liquid nitrogen, and stored at –80°C. Colons were also collected aseptically and the intestinal contents were excluded from the intestine with PBS. Then, tissues were stored at –80°C for gene expression measurement.

To examine the role of the gut microbiota on acute leptospirosis, the gut microbiota depletion and fecal microbiota transplantation (FMT: a therapeutic procedure aimed at restoring a normal intestinal microbiota by application of fecal microorganisms from a healthy subject into the gastrointestinal tract of a dysbiotic subject [*Schmidt et al., 2022*]) were performed as described previously with some modifications (*Sun et al., 2019*). Briefly, 6-week-old female hamsters were administered antibiotics (ampicillin, 100 mg/kg; metronidazole, 100 mg/kg; neomycin sulfate, 100 mg/kg [Sigma-Aldrich, USA]; and vancomycin, 50 mg/kg [BiochemPartner, China]) intragastrically once daily for 10 consecutive days (*Zhai et al., 2023*). Fresh fecal pellets were collected from uninfected female hamsters and then resuspended in sterile normal PBS (one fecal pellet in 1 ml of PBS). The pellets were immediately homogenized and the homogenate was centrifuged at 100 rpm, 4°C for 5 min. 200 µl of the supernatant was given to the gut microbiota-depleted hamster by oral gavage for five consecutive days after antibiotic treatment was stopped. Then, hamsters were intraperitoneally infected with $10^6$ *L. interrogans*. Hamsters were monitored daily for 21 d to record the survival rate. Hamsters were euthanized when they were moribund. Blood was collected to determine leptospiral load and the gene expression of inflammatory cytokines at 6 d p.i.

To explore the effect of LPS neutralization on severe leptospirosis, 6-week-old male or female hamsters were inoculated intraperitoneally with $10^7$ leptospires. Treatments were started immediately after detection of the first death regardless of the assigned group (*Soares et al., 2014*). Group 1: saline control (400 µl/hamster, i.p.); group 2: polymyxin B (PMB) (1 mg/kg, i.p., Solarbio, China); group 3: antibody against *Leptospira* (Ab) (16 mg/kg, subcutaneous injection); group 4: doxycycline (Dox) (5 mg/kg, i.p., Solarbio); group 5: PMB (1 mg/kg, i.p.) and Ab (16 mg/kg, subcutaneous injection); group 6: PMB (1 mg/kg, i.p.) and Dox (5 mg/kg, i.p.). Hamsters were treated twice a day for three consecutive days. Hamsters were monitored daily for 21 d. Hamsters were euthanized when they were

moribund. Blood from female hamsters was collected to determine the leptospiral load and gene expression of inflammatory cytokines at 6 d p.i.

To explore the effect of LPS neutralization on another leptospiral serotype infection, 6-week-old female hamsters were inoculated intraperitoneally with lethal $10^6$ leptospires (56606), of which the infective doses were determined by a previous study (*Jin et al., 2016*). Treatments were started immediately after detection of the first death regardless of the assigned group. Group 1: saline control (400 μl/hamster, i.p.); group 2: PMB (1 mg/kg, i.p., Solarbio); group 3: Dox (5 mg/kg, i.p., Solarbio); group 4: PMB (1 mg/kg, i.p.) and Dox (5 mg/kg, i.p.). Hamsters were treated twice a day for three consecutive days. Hamsters were monitored daily for 21 d. Hamsters were euthanized when they were moribund.

## Part II: Rabbit infection

The purification of polyclonal antibody (Ab) was performed as previously described (*Jin et al., 2016*). Briefly, six 8-month-old female New Zealand White rabbits were injected intravenously into the marginal ear vein with a dose of $2 \pm 4 \times 10^8$ live leptospires/ml in PBS according to the following schedule: day 1, 1 ml; day 6, 2 ml; day 11, 4 ml; and days 16 and 21, 6 ml each. One week after the last injection, the rabbits were anesthetized and then exsanguinated through cardiocentesis. The rabbits were then euthanized. During the immunization period, the rabbits did not exhibit clinical signs of leptospirosis. Antisera were collected by centrifuging blood at 3000 × *g*. The polyclonal Ab IgG was purified from the antisera using the caprylic acid ammonium sulfate precipitation method from the antisera. The concentration of the IgG-polyclonal Ab was determined with the BCATM Protein Assay Kit (Thermo, USA) and adjusted to a final concentration of 20 mg/ml with PBS. The Ab was stored at –20°C until used for treatment. The protocol was approved by the Committee on the Ethics of Animal Experiments of the First Norman Bethune Hospital of Jilin University, China [(2013) clinical trial (2013-121)] (*Jin et al., 2016*).

## Bacterial isolation and culture

For isolation of the translocated microbiota, 6-week-old female hamsters were intraperitoneally infected with $10^7$ leptospires. Hamsters were anesthetized by intraperitoneal injection of a mix of ketamine/xylazine (200 and 10 mg/kg, respectively). Blood from the uninfected group and AM group by cardiac puncture was plated on LB or MacConkey agar plates aerobically or anaerobically at 37°C for 36 hr.

## In vivo barrier permeability

Six-week-old female hamsters were intraperitoneally infected with $10^7$ leptospires. The permeability of the intestine was determined at 0 d, 2 d, and AM p.i. Hamsters were starved for 6 hr before gavage with FITC-dextran (4 kDa, 400 mg/kg body/hamster). After 2 hr, hamsters were anesthetized by intraperitoneal injection of a mix of ketamine/xylazine (200 and 10 mg/kg, respectively) and blood was collected and centrifuged at 3000 rpm and 4°C for 10 min. The samples were analyzed with a fluorescence spectrophotometer (Synergy II plate reader with Gen5 software; BioTek Instruments, Winooski, VT) (*Spalinger et al., 2019*).

## Minimum inhibitory concentration determination

The minimum inhibitory concentration (MIC) of PMB and Dox against *L. interrogans* serovar Lai strain Lai 56601 was determined according to a previous method (*Murray and Hospenthal, 2004*). Briefly, leptospires in the exponential phase were deposited at a final concentration of $2 \times 10^6$ leptospires/ml in each well of 96-well plates with serial twofold dilutions of PMB and Dox ranging from 32 to 0.016 μg/l in EMJH. The final volume was 200 μl in each well. The plates were incubated for 3 d at 30°C, and then 20 μl of Alamar Blue (Invitrogen, Thermo Fisher Scientific) was added to each well. Then, the samples were incubated at 30°C for 2 d. Each strain–drug combination was tested in duplicate, and positive (bacteria and no antibiotic added) and negative (no bacteria added) controls were included in each plate. The results were recorded using a microplate reader (MultiSkan FC, Thermo Fisher Scientific).

## Limulus amebocyte lysate assay

Blood from uninfected hamsters and AM hamsters was collected and centrifuged at 3000 rpm for 10 min. Then, serum was diluted (1:10) and detected by the Limulus amebocyte lysate assay according

to the manufacturer's instructions (LAL Chromogenic Endotoxin Quantitation Kit; Xiamen Biotechnology Co., Ltd, China). *E. coli* LPS at 1 EU/ml and *E. coli* LPS at 10 EU/ml (Sigma-Aldrich) were used as positive controls.

## Fluorescence in situ hybridization

Six-week-old female hamsters were injected intraperitoneally with $10^7$ leptospires. Hamsters were euthanized at 0 d, 2 d, and AM p.i. Colons were collected aseptically. The intestinal contents were excluded from the intestine with PBS and a segment of the colon was fixed in 4% paraformaldehyde solution overnight, washed and passed through 15 and 30% sucrose solutions. The sample was then embedded in optimal cutting temperature compound (O.C.T., Tissue-Tek) and cryo-sectioned into 5 μm longitudinal sections (Leica). Slides were equilibrated in hybridization buffer (0.9 M NaCl, 20 mM Tris–HCl, 0.01% sodium dodecyl sulfate, 10% formamide, pH 7.5) and incubated in Cy3-labeled 10 ng/μl FISH probe (Comate Bioscience Co., Ltd, China) for 14 hr at 42°C in a humidified chamber. The probe information is listed in *Supplementary file 1*. Slides were then incubated for 20 min in wash buffer (0.9 M NaCl, 20 mM Tris–HCl, pH 7.5) preheated to 42°C and washed three times. Samples were then incubated in the dark with 10 μg/ml DAPI (Servicebio Technology Co., Ltd, China) in PBS for 10 min at room temperature, washed three times with PBS, and mounted with Vectashield mounting medium (Vector Labs). Images were acquired on a Nikon A1 confocal microscope.

## FISH and IF double labeling (FISH/IF)

After the preparation and digestion of paraffin-embedded sections and before hybridization, the probe hybridization solution was added at a concentration of 1 μM and the sections were incubated in a humidified chamber to hybridize with the Cy3-labeled 10 ng/μl FISH probe (Comate Bioscience Co., Ltd) for 14 hr at 42°C. Then, the sections were washed, blocked, and incubated with rabbit anti-*L. interrogans* serovar Lai strain Lai 56601 (1:500) as the primary antibody overnight at 4°C. The next day, the samples were washed with PBS three times for 5 min. After washing, the samples were blocked with 5% BSA and incubated at room temperature for 1 hr with the secondary antibody. Samples were then incubated in the dark with 10 μg/ml DAPI (Servicebio Technology Co., Ltd) in PBS for 10 min at room temperature, washed three times with PBS, and mounted with Vectashield mounting medium (Vector Labs) (*Xu et al., 2023*; *Wu et al., 2023*). Images were acquired on a Nikon A1 confocal microscope.

## Isolation of primary macrophages

Primary macrophages were isolated as previously described (*Xie et al., 2022*). Six-week-old female hamsters were injected intraperitoneally with 2 ml of 3% thioglycolate dissolved in distilled deionized water. Hamsters were euthanized after 4 d, and macrophages were lavage by sterile PBS and enriched by plating the lavage cells on tissue culture plates in RPMI 1640 supplemented with 10% FCS and 1% penicillin and streptomycin at 37°C for 2 hr. Then, the plate was washed three times with sterile PBS to remove nonadherent cells.

## Culture of primary macrophages with fecal filtration or serum

$1 \times 10^6$ cells/well were seeded in a 12-well plate. Fecal filtration or serum was collected from uninfected hamsters and AM hamsters. Fecal pellets were homogenized using PBS (1 ml per 0.1 g of fecal matter). The homogenate was centrifuged at 100 rpm and 4°C for 5 min and filtered using 0.22 μm syringe filters. The quantity of the fecal filtration or serum was adjusted to 5 or 20% of the total volume to stimulate macrophages in vitro. After incubation at 37°C for 4 hr, RNA was extracted with TRIzol.

To determine the role of LPS in the induction of cytokine release, fecal filtration or serum was preincubated with LPS neutralizing reagent PMB (Solarbio) for 2 hr at 37°C, resulting in a final concentration of 50 μg/ml PMB after addition to the macrophages. After 4 hr of incubation at 37°C, RNA was extracted with TRIzol.

## Real-time and reverse transcription-qPCR (RT-qPCR)

Total RNA of the colon samples was extracted with TRIzol (Invitrogen, USA). RNA was reverse-transcribed into cDNA by using random primers from a TransScript One-Step gDNA Removal kit and cDNA Synthesis SuperMix (TransGen Biotech, China). The primers used in this study are listed

in *Supplementary file 1*. The PCR conditions were as follows: 50°C for 2 min, 95°C for 10 min, followed by 45 cycles of amplification at 95°C for 15 s and 60°C for 60 s (*Cao et al., 2018*). qPCR was conducted using an Applied Bioscience 7500 thermocycler and FastStart Universal SYBR Green Master Mix (Roche Applied Science, Germany). The expression of target genes was normalized to that of GAPDH using the $2^{-\Delta\Delta CT}$ method.

## Bacterial load and qPCR assay

The leptospiral burdens in organs were determined by quantitative PCR using an Applied Bioscience 7500 thermocycler and FastStart Universal SYBR green Master (Roche Applied Science). The specimens (0.09–0.15 g) of tissues were homogenized with PBS (w/v, 1/10). The homogenate was centrifuged at 2000 rpm and 4°C for 5 min. The supernatant was transferred into a new tube. Then, it was centrifuged at 12,000 rpm and 4°C for 5 min. The pellets were extracted using the TIANamp Bacteria DNA kit (Tiangen, China) according to the manufacturer's instructions (*Zhang et al., 2020*). The concentration of DNA was measured by spectrometry. The genomic DNA of a counted number of *L. interrogans* was used as a calibrator. The primers specific for *LipL32* were used to detect leptospires (forward primer, 5′-TCGCTGAAATRGGWGTTCGT-3′; reverse primer, 5′-CGCCTGGYTCMCCGATT-3′). The leptospiral load was presented as the number of genome equivalents per μg of tissue DNA (*Zhang et al., 2016*).

## Histopathological examination

Colons from the uninfected group and AM group were collected and fixed in buffered 4% formaldehyde. Hematoxylin and eosin (H&E) staining was performed on 4–5 μm paraffin-embedded sections. Blinded samples were then scored by two pathologists. Each section was evaluated for inflammation, epithelial hyperplasia, erosion and ulceration, and the extent of lesion as listed in *Supplementary file 2*; *Sanchez et al., 2018*.

## Microbiota assay

Fresh fecal pellets were collected under sterile conditions, immediately frozen in liquid nitrogen, and then stored at −80°C. Total genomic DNA from the samples was extracted using the CTAB/SDS method. DNA concentration and purity were monitored on 1% agarose gels. Then, 16S rRNA genes of distinct regions were amplified using specific primers with barcodes (27F, AGAGTTTGATCMTGGCTCAG; 1492R, ACCTTGTTACGACTT). All PCRs were carried out with TransStart FastPfu DNA Polymerase (TransGen Biotech). PCR products were purified using the QIAGEN Gel Extraction Kit (QIAGEN, Germany). Sequencing libraries were generated using the SMRTbell Template Prep Kit (PacBio) following the manufacturer's recommendations. The library quality was assessed on the Qubit 2.0 Fluorometer (Thermo Scientific) and FEMTO Pulse system. Finally, the library was sequenced on the PacBio Sequel platform. Raw sequences were initially processed through the PacBio SMRT portal. Sequences were filtered for a minimum of three passes, and a minimum predicted accuracy of 90% (minfullpass = 3, minPredictedAccuacy = 0.9). The predicted accuracy was 90%, which was defined as the threshold below which CCS was considered noise. The files generated by the PacBio platform were then used for amplicon size trimming to remove sequences outside the expected amplicon size (minLength 1340 bp, maxLength 1640 bp). The reads were assigned to samples based on their unique barcode and trimmed by removing the barcode and primer sequence. The reads were compared with the reference database using the UCHIME algorithm to detect chimeric sequences, and then the chimeric sequences were removed to obtain clean reads. Alpha diversity (Chao1 and Shannon) was calculated with QIIME (version 1.9.1) and displayed with R software (version 2.15.3). PCoA was performed to obtain principal coordinates and visualize the complex, multidimensional data. A distance matrix of weighted UniFrac among samples obtained before was transformed to a new set of orthogonal axes by which the maximum variation factor was demonstrated by the first principal coordinate, and the second maximum variation factor was demonstrated by the second principal coordinate. PCoA results were displayed using the WGCNA package, stat packages, and ggplot2 package in R software (version 2.15.3).

## Statistical analysis

Survival differences between the study groups were compared by using the Kaplan–Meier log-rank test. All values are expressed as the mean ± SEM. Differences between mean values of normally

distributed data were analyzed using the Wilcoxon rank-sum test. The results were considered statistically significant at $p < 0.05$.

## Acknowledgements

We thank Dr. Xiaokui Guo (Shanghai Jiao Tong University, Shanghai, China) for providing 56601 and 56606. This work was supported by the National Natural Science Foundation of China (nos. 32172872 and 32302879), the Fundamental Research Funds for the Central Universities, China Postdoctoral Science Foundation-funded project (BX20220131), and Basic and Applied Basic Research Foundation of Guangdong Province (2022A1515110966).

## Additional information

### Funding

| Funder | Grant reference number | Author |
|---|---|---|
| National Natural Science Foundation of China | 32172872 | Yongguo Cao |
| National Natural Science Foundation of China | 32302879 | Jiuxi Liu |
| China Postdoctoral Science Foundation | BX20220131 | Xufeng Xie |
| Basic and Applied Basic Research Foundation of Guangdong Province | 2022A1515110966 | Jiuxi Liu |

The funders had no role in study design, data collection and interpretation, or the decision to submit the work for publication.

### Author contributions

Xufeng Xie, Conceptualization, Data curation, Formal analysis, Funding acquisition, Writing - original draft, Project administration; Xi Chen, Software, Investigation, Methodology; Shilei Zhang, Software, Formal analysis, Investigation, Methodology; Jiuxi Liu, Conceptualization, Funding acquisition, Investigation; Wenlong Zhang, Yongguo Cao, Conceptualization, Funding acquisition, Writing – review and editing

### Author ORCIDs

Yongguo Cao http://orcid.org/0000-0002-9533-7516

### Ethics

All animal experiments were conducted according to the regulations of the Administration of Affairs Concerning Experimental Animals in China. The protocol was approved by the Institutional Animal Care and Use Committee of Jilin University (20170318). All surgery was performed under sodium pentobarbital anesthesia, and every effort was made to minimize suffering.

Reviewer #1 (Public Review) https://doi.org/10.7554/eLife.10.7554/eLife.96065.3.2.sa1
Reviewer #2 (Public Review): https://doi.org/10.7554/eLife.10.7554/eLife.96065.3.2.sa2
Reviewer #3 (Public Review): https://doi.org/10.7554/eLife.10.7554/eLife.96065.3.2.sa3
Author response https://doi.org/10.7554/eLife.10.7554/eLife.96065.3.2.sa4

## Additional files

### Supplementary files

- Supplementary file 1. Primers and probes used in the article.
- Supplementary file 2. Histological colitis scoring system.

• MDAR checklist

## Data availability

Sequencing data have been deposited in NCBI BioProject under accession codes PRJNA772361 and PRJNA1036417. All data generated or analysed during this study are included in the manuscript and supporting files; source data files have been provided for Figure 1 to Figure 7 and Figure 1—figure supplement 1, Figure 1—figure supplement 2, Figure 6—figure supplement 1, Figure 6—figure supplement 2, Figure 7—figure supplement 1, and Figure 7—figure supplement 2. Figure 1—source data 1, Figure 1—figure supplement 1—source data 1, Figure 1—figure supplement 2—source data 1, Figure 2—source data 1, Figure 3—source data 1, Figure 4—source data 1, Figure 5—source data 1, Figure 6—source data 1, Figure 6—figure supplement 1—source data 1, Figure 6—figure supplement 2—source data 1, Figure 7—source data 1, Figure 7—figure supplement 1—source data 1, and Figure 7—figure supplement 2—source data 1 contain the numerical data used to generate the figures.

The following datasets were generated:

| Author(s) | Year | Dataset title | Dataset URL | Database and Identifier |
|---|---|---|---|---|
| Xie X, Chen X, Zhang S, Liu J, Cao Y | 2021 | The role of microbiota during leptospira infection | https://www.ncbi.nlm.nih.gov/bioproject/PRJNA772361 | NCBI BioProject, PRJNA772361 |
| Xie X, Chen X, Zhang S, Liu J, Zhang W, Cao Y | 2023 | The role of Abx treatment and FMT on the microbiota of hamster | https://www.ncbi.nlm.nih.gov/bioproject/PRJNA1036417 | NCBI BioProject, PRJNA1036417 |

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
