## [Editor Report · eLife assessment]

The gut microbiota influences many infectious diseases; however, its role in *Leptospirosis* remains unclear. In this **fundamental** work, Xie et al. use a hamster model to show that *Leptospira* infection leads to gut pathology, an altered gut microbiota, and increased translocation. A combined use of antibiotics and LPS neutralization prolonged survival, providing a potential new therapeutic approach. This study utilizes **compelling** methods to provide new insights into this emerging disease, which could be dissected further in future studies aimed at gaining mechanistic insight and assessing the translational relevance of these discoveries.

---

## [Referee Report · Reviewer #1 (Public Review)]

Summary:

In this study, Xie and colleagues aimed to explore the function and potential mechanisms of the gut microbiota in a hamster model of severe leptospirosis. The results demonstrated that Leptospira infection was able to cause intestine damage and inflammation. Leptospira infection promoted an expansion of Proteobacteria, increased gut barrier permeability, and elevated LPS levels in the serum. Thus, they proposed an LPS-neutralization therapy which improved the survival rate of moribund hamsters combined with antibody therapy or antibiotic therapy.

Strengths:

The work is well-designed and the story are interesting to me. The gut microbiota is essential for immunity and systemic health. Many life-threatening pathogens, such as SARS-CoV-2 and other gut-damaged infection, have the potential to disrupt the gut microbiota in the later stages of infection, causing some harmful gut microbiota-derived substances to enter the bloodstream. It is emphasized that in addition to exogenous pathogenic pathogens, harmful substances of intestinal origin should also be considered in critically ill patients.

---

## [Referee Report · Reviewer #2 (Public Review)]

Severe leptospirosis in humans and some mammals often meet death in the endpoint. In this article, authors explored the role of the gut microbiota in severe leptospirosis. They found that Leptospira infection promoted a dysbiotic gut microbiota with an expansion of Proteobacteria and LPS neutralization therapy synergized with antileptospiral therapy significantly improved the survival rates in severe leptospirosis. This study is well-organized and has potentially important clinical implications not only for severe leptospirosis but also for other gut-damaged infections.

---

## [Referee Report · Reviewer #3 (Public Review)]

Summary:

This is a well prepared manuscript which presented interesting research result.

Strengths:

The omics method produced unbiased results.

Weaknesses:

LPS neutralization is not new method for treating leptospiral infection.

---

## [Author Response]

The following is the authors’ response to the original reviews.

**Reviewer #1 (Comments to the Author):**
Summary:In this study, Xie and colleagues aimed to explore the function and potential mechanisms of the gut microbiota in a hamster model of severe leptospirosis. The results demonstrated that Leptospira infection was able to cause intestine damage and inflammation. Leptospira infection promoted an expansion of Proteobacteria, increased gut barrier permeability, and elevated LPS levels in the serum. Thus, they proposed an LPS-neutralization therapy which improved the survival rate of moribund hamsters combined with antibody therapy or antibiotic therapy.Strengths:The work is well-designed and the story is interesting to me. The gut microbiota is essential for immunity and systemic health. Many life-threatening pathogens, such as SARS-CoV-2 and other gut-damaged infection, have the potential to disrupt the gut microbiota in the later stages of infection, causing some harmful gut microbiota-derived substances to enter the bloodstream. It is emphasized that in addition to exogenous pathogenic pathogens, harmful substances of intestinal origin should also be considered in critically ill patients.Weaknesses:Q1: There are many serotypes of Leptospira, it is suggested to test another pathogenic serotype of Leptospira to validate the proposed therapy.

That’s a constructive suggestion. We have tested another pathogenic serotype of Leptospira (L. interrogans serovar Autumnalis strain 56606) to verify the LPS-neutralization therapy combined with antibiotic therapy (Supplementary Fig. S9B). The results showed that the combination of the LPS-neutralization therapy with antibody therapy or antibiotic therapy also significantly improved the survival rate of hamsters infected by 56606.

Q2: Authors should explain why the infective doses of leptospires was not consistent in different study.

Thank you for your comment. To examine the role of the gut microbiota on acute leptospirosis, the infective doses of leptospires was chosen for 106, while in other sections of the study, the infective doses of leptospires was chosen for 107. In fact, we also used 107 leptospires to infect hamsters, however, the infective doses of 107 leptospires might be overdose, there was no significant difference on the survival rate between the control group and the Abx-treated group. A previous study also highlighted that the infective doses of leptospires was important in the investigating the sex on leptospirosis, as male hamsters infected with L. interrogans are more susceptible to severe leptospirosis after exposure to lower infectious doses than females (103 leptospires but not 104 leptospires) (1).

Reference

(1) GOMES C K, GUEDES M, POTULA H H, et al. Sex Matters: Male Hamsters Are More Susceptible to Lethal Infection with Lower Doses of Pathogenic Leptospira than Female Hamsters (J). Infect Immun, 2018, 86(10).

Q3: In the discussion section, it is better to supplement the discussion of the potential link between the natural route of infection and leptospirosis.

Thank for your suggestion. We have supplemented it in the discussion (line 523-527 in the track change PDF version).

Q4: Line 231, what is the solvent of thioglycolate?

We have supplemented it in the manuscript (line 242-243 in the track change PDF version).

Q5: Lines 962-964, there are some mistakes which are not matched to Figure 7.

Thank you for pointing that out, we have corrected it in the manuscript.

**Reviewer #2 (Comments to the Author):**
Summary:Severe leptospirosis in humans and some mammals often meet death in the endpoint. In this article, authors explored the role of the gut microbiota in severe leptospirosis. They found that Leptospira infection promoted a dysbiotic gut microbiota with an expansion of Proteobacteria and LPS neutralization therapy synergized with antileptospiral therapy significantly improved the survival rates in severe leptospirosis. This study is well-organized and has potentially important clinical implications not only for severe leptospirosis but also for other gut-damaged infections.Weaknesses:Q1: In the Introduction section and Discussion section, the authors should describe and discuss more about the differences in the effect of Leptospira infection between mice and hamsters, so that the readers can follow this study better.

Thank you for your suggestion, we have supplemented it in the manuscript (line 62-66 in the track change PDF version).

Q2: Lines 92-95, the authors should explain why they chose two different routines of infection.

Thank you for your comment, we have explained it in the manuscript (line 100 in the track change PDF version).

Q3: Line 179-180, the concentration of PMB and Dox is missed, and 0.016 μg/L is just ok.

We have corrected it in the manuscript.

Q4: "μL" or "μl" and "mL" or "ml' should be uniform in the manuscript.

Thank you for your suggestion, we have revised it in the manuscript.

Q5: In the culture of primary macrophages, how many cells are inoculated in the plates should be described clearly.

We have supplemented it in the manuscript (line 250 in the track change PDF version).

Q6: Line 271, it is better to list primers used for leptospiral detection in the text. Because it allows readers to find the information they need more directly.

Thank you for your suggestions, we have supplemented it in the manuscript (line 281-284 in the track change PDF version).

Q7: Line 366-369, Lactobacillus seems to be a kind of key bacteria during Leptospira infection. A previous study (doi: 10.1371/journal.pntd.0005870) also demonstrated that pre-treatment with Lactobacillus plantarum prevented severe pathogenesis in mice. The authors should discuss the potential probiotic for leptospirosis prevention.

We have discussed it in the manuscript (line 564-566 in the track change PDF version).

Q8: Lines 450-451, not all concentrations of fecal filtration from two groups upregulated all gene expression mentioned in the text, the authors should correct it.

Thank you for pointing that out, we have corrected it in the manuscript (line 461-462 in the track change PDF version).

**Reviewer #3 (Comments to the Author):**
Summary:This is a well-prepared manuscript that presented interesting research results. The only defect is that the authors should further revise the English language.Strengths:The omics method produced unbiased results.Weaknesses:Q1: LPS neutralization is not a new method for treating leptospiral infection.

Thank you for your comment. Yes, LPS neutralization is not a new method for treating leptospiral infection, most of which might focus on leptospiral LPS. In addition, Leptospira seemed to be naturally resistant to polymyxin B (1). Recently, neutralizing gut-derived LPS was applied in other diseases which significantly relieved diseases (2-3). In this study, we found that Leptospira infection promoted an expansion of Proteobacteria, increased gut barrier permeability, and elevated LPS levels in the serum. Thus, we proposed an LPS-neutralization therapy which improved the survival rate of moribund hamsters combined with antibody therapy or antibiotic therapy.

Reference

(1) LIEGEON G, DELORY T, PICARDEAU M. Antibiotic susceptibilities of livestock isolates of leptospira (J). Int J Antimicrob Agents, 2018, 51(5):693-699.

(2) MUNOZ L, BORRERO M J, UBEDA M, et al. Intestinal Immune Dysregulation Driven by Dysbiosis Promotes Barrier Disruption and Bacterial Translocation in Rats With Cirrhosis (J). Hepatology, 2019, 70(3):925-938.

(3) ZHANG X, LIU H, HASHIMOTO K, et al. The gut-liver axis in sepsis: interaction mechanisms and therapeutic potential (J). Crit Care, 2022, 26(1):213.

Q2: The authors should further revise the English language used in the text.

Thank you for your suggestion, our manuscript has been polished by American Journal Experts (certificate number: 81C8-C5C1-9D5D-109D-3F23).